# Evaluation of the Safety, Tolerability and Immunogenicity of ShigETEC, an Oral Live Attenuated *Shigella*-ETEC Vaccine in Placebo-Controlled Randomized Phase 1 Trial

**DOI:** 10.3390/vaccines10020340

**Published:** 2022-02-21

**Authors:** Petra Girardi, Shushan Harutyunyan, Irene Neuhauser, Katharina Glaninger, Orsolya Korda, Gábor Nagy, Eszter Nagy, Valéria Szijártó, Denes Pall, Krisztina Szarka, Gábor Kardos, Tamás Henics, Frank J. Malinoski

**Affiliations:** 1Eveliqure Biotechnologies GmbH, Karl-Farkas-Gasse 22, 1030 Vienna, Austria; shushan.harutyunyan@eveliqure.com (S.H.); irene.neuhauser@eveliqure.com (I.N.); katharina.glaninger@eveliqure.com (K.G.); orsolya.korda@eveliqure.com (O.K.); gabor.nagy@eveliqure.com (G.N.); eszter.nagy@eveliqure.com (E.N.); tamas.henics@eveliqure.com (T.H.); frank.malinoski@eveliqure.com (F.J.M.); 2CEBINA GmbH, Karl-Farkas-Gasse 22, 1030 Vienna, Austria; valeria.szijarto@eveliqure.com; 3Department of Medicine, Division of Clinical Pharmacology, University of Debrecen, Nagyerdei krt. 98, H-4032 Debrecen, Hungary; pall.denes@unideb.hu; 4Department of Medical Microbiology, Faculty of Medicine, University of Debrecen, Nagyerdei krt. 98, H-4032 Debrecen, Hungary; szkrisz@med.unideb.hu (K.S.); kg@med.unideb.hu (G.K.); 5Department of Metagenomics, University of Debrecen, Nagyerdei krt. 98, H-4032 Debrecen, Hungary

**Keywords:** *Shigella*, enterotoxigenic *E. coli*, vaccine, oral, Phase 1 clinical study, ETEC toxins

## Abstract

Background: *Shigella* spp. and enterotoxigenic *Escherichia coli* (ETEC) cause high morbidity and mortality worldwide, yet no licensed vaccines are available to prevent corresponding infections. A live attenuated non-invasive *Shigella* vaccine strain lacking LPS O-antigen and expressing the ETEC toxoids, named ShigETEC was characterized previously in non-clinical studies. Methods: ShigETEC was evaluated in a two-staged, randomized, double-blind and placebo-controlled Phase I clinical trial. A single dose of increasing amounts of the vaccine was given to determine the maximum tolerated dose and increasing number of immunizations were administered with an interval based on the duration of shedding observed. Results: Oral immunization with ShigETEC was well tolerated and safe up to 4-time dosing with 5 × 10^10^ colony forming units. ShigETEC induced robust systemic immune responses against the *Shigella* vaccine strain, with IgA serum antibody dominance, as well as mucosal antibody responses evidenced by specific IgA in stool samples and in ALS (Antibodies in Lymphocyte Supernatant). Anti- ETEC toxin responses were detected primarily in the 4-times immunized cohort and for the heat-labile toxin correlated with neutralizing capacity. Conclusion: ShigETEC is a promising vaccine candidate that is scheduled for further testing in controlled human challenge studies for efficacy as well as in children in endemic setting for safety and immunogenicity.

## 1. Introduction

*Shigella* spp. and enterotoxigenic *Escherichia coli* (ETEC) are leading causes of enteral diseases among children living in endemic countries with poor hygienic conditions [1,2]. An annual 200 million episodes of ETEC and *Shigella* diarrhea are estimated in children under the age of 5 [3]. The incidence in the adult endemic population decreases due to immunity gained through repeated exposure to these bacteria. On the other hand, travelers and deployed military personnel with no pre-existing immunity are vulnerable to these pathogens. Approximately 25–50% of the annual >100 million travelers’ diarrhea cases can be attributed to these two bacterial pathogens [4,5]. Although mortality rates from intestinal infections have decreased recently, *Shigella* spp. and ETEC were reported to be responsible for an estimated >210,000 and >50,000 deaths, respectively, in 2016 worldwide [1]. Despite of the high morbidity and mortality due to these pathogens, no approved vaccine against *Shigella* or ETEC is currently available.

*Shigella* is an invasive enteral pathogen. It traverses the mucosal barrier via M cells, and following induction of pyroptosis in the underlying macrophages, it infects enterocytes from the basolateral side. After intracellular replication, *Shigella* spreads to adjacent enterocytes laterally. The consequent cell death and proinflammatory cytokine cascade cause an influx of neutrophil granulocytes that trigger massive inflammation and tissue destruction, the hallmarks of dysentery [6]. Besides, certain serogroups (e.g., *S. flexneri* 2) express enterotoxins, which may be responsible for watery diarrhea [7]. *Shigella* (both natural infection and vaccines) is considered to induce serotype specific immunity, based on the structurally diverse lipopolysaccharide (LPS) O-antigens. In fact, the immune response is dominated by antibodies against O-antigens, which is considered as a correlate of protection [8]. To achieve broad protection against the total ~50 serotypes of the 4 *Shigella* species, multivalent vaccines are required. Vaccine candidates developed in the last >50 years include killed whole cell, live attenuated, and subunit vaccines, several of them reached clinical phase of testing [9]. Among these different classes, live attenuated strains appear to have several advantages in terms of vaccination route, production cost, and most importantly for best mimicking the natural infection and to induce protective mucosal immune responses. The major challenge in developing live vaccine strains is to find the right balance between immunogenicity and safety. Reactogenicity remains a major hurdle that necessitates lower vaccine doses, which often do not induce sufficient immunity and consequently protection [10].

In contrast to *Shigella*, ETEC is a non-invasive pathogen that colonizes the small intestinal mucosa by adhesion with multiple heterogenous fimbriae and produces at least one of the two exotoxins: the heat-labile (LT) and heat-stable (ST) enterotoxins. Both toxins bind to their corresponding receptors on the apical surface of enterocytes and induce massive fluid secretion to the gut lumen resulting in watery diarrhea [11]. Vaccine development against ETEC is hindered by the heterogenous nature of the fimbrial adhesins as well as the inherent low immunogenicity of ST (a short peptide). An ideal vaccine candidate against ETEC should induce neutralizing mucosal immunity against the majority of the most common fimbrial types and/or both enterotoxins [12].

A combined *Shigella*/ETEC live vaccine candidate, termed ShigETEC, has been described recently [13]. The parental strain (*S. flexneri* 2a 2457T) has been attenuated through 3 independent genetic deletions-Δ*ipaBC*, Δ*rfbF*, Δ*setBA*-resulting in a non-invasive, rough vaccine strain, which is unable to secrete *Shigella* exotoxins. ShigETEC expresses a fusion protein of the B subunit of LT (LTB) and mutated ST(N12S) from the large invasion plasmid. Furthermore, the essential gene *infA* has been trans-positioned from the chromosome to the invasion plasmid, in order to stabilize the plasmid and thereby ensure expression of the heterologous ETEC antigens as well as that of plasmid/encoded *Shigella* antigens. It has been envisioned that the non-invasive nature of the vaccine strain allows immunizations using high oral doses with good safety profile. The lack of LPS O-antigens of the vaccine strain allows higher immunogenicity of shared conserved *Shigella* antigens that may be responsible for the serotype independent protection observed in animal models [13].

To assess the safety and immunogenicity of oral vaccination with ShigETEC, a first-in-man study was performed in human volunteers.

## 2. Materials and Methods

This Phase 1 study was conducted at the University of Debrecen, Hungary and was approved by The National Institute of Pharmacy and Nutrition Hungary (OGYÉI/13385-12/2020), and under the Authorization of Medical Research Council, Ethics Committee for Clinical Pharmacology (IV/2175-0/2020-EKL) and was registered under EudraCT: 2020-000248-79.

Study design: This Phase 1 study was designed as a double-blinded, randomized, placebo-controlled, dose-escalating trial, conducted in two stages (depicted in Figure 1). In Stage 1, 48 subjects (2 vaccine recipients to each placebo recipient) were enrolled sequentially in 4 ascending dose groups (12 subjects per group) to receive a single oral dose of the ShigETEC vaccine starting at a dose of 1 × 10^9^ CFU (Cohort 1A), followed by 1 × 10^10^ CFU (Cohort 1B), 5 × 10^10^ CFU (Cohort 1C) and 2 × 10^11^ CFU (Cohort 1D). Two sentinel subjects (one vaccine and one placebo) were enrolled for each dose group prior to the enrolment of the remaining subjects in that group. Provided that no stopping rules (any serious adverse event attributed to the vaccine or evidence of moderate to severe shigellosis as reviewed by the Safety Review Committee) were met through 6 days of follow-up for the sentinel subjects, the remainder of the subjects for that group were enrolled and dosed. Stage 1 was conducted in an inpatient hospital setting through day 6. Subjects in each group asymptomatic for shigellosis-like illness were discharged to outpatient follow-up 6 days after. Any subject with symptomatic illness were scheduled to be treated with an appropriate course of antibiotic at the time of discharge. Stool samples (as available with the target of daily) were tested for the presence of the ShigETEC vaccine strain (by both PCR and culture as described below) until at least 2 consecutive specimens were negative. If shedding persisted through 14 days, then the participant would be treated with antibiotics regardless of any symptomatology. All subjects were followed through 60 days following immunization to collect information on adverse events and collect samples for immunogenicity analysis. Stage 2 of the study were performed in an outpatient setting and enrolled 36 subjects (2 vaccine recipients to each placebo recipient) in 3 groups (12 subjects per group) sequentially receiving 2 (Cohort 2A), 3 (Cohort 2B) or 4 (Cohort 2C) vaccinations, respectively. The vaccine dose and immunization interval were based on data from Stage 1 with the goal to dose with the maximum tolerated dose at or near the end of the shedding cycle. The progression between each of the dose regimens depended on satisfactory safety data of the prior dose regimen using the same safety parameters that were used in Stage 1. Any concomitant medication, including antibiotics, administered/or received during the trial was recorded in the case report forms. Inclusion and exclusion criteria are listed in Appendix A. Human leukocyte antigen (HLA) B27 positive individuals were excluded due to the potential higher risk for reactive arthritis following gastrointestinal infection with Gram-negative bacteria [14].

Participant recruitment: Potential volunteers were recruited from the general population of the clinical trial site area using Ethics Committee approved advertisement mechanisms. Subjects were healthy adult male and non-pregnant females between 18 and 45 years of age with no documented history of *Shigella* or ETEC disease. Initial screening of volunteers was conducted based on questions related to travel history, age, medical history, vaccination history, type of employment, and availability for the study duration. Subjects meeting the eligibility requirements and who agreed to continue moved forward to informed consent to allow further evaluation, including further interview questions, collection and analysis of blood samples, and a directed medical examination as needed to define inclusion/exclusion criteria for qualifying subjects for randomization. The randomization was done for one clinical site, and two different study treatments (vaccine and placebo). The randomization was performed separately for the two stages of the study, and for the sentinel participants and the other participants in Stage 1, and separately for all treatment groups. Randomization was performed using the nQuery Advisor 6.01 software and sequential randomization numbers were only given to participants who met the inclusion/exclusion criteria, were enrolled to the study and were present at the immunization. Randomization was done only right before immunization.

Vaccine manufacturing, preparation, and administration: the ShigETEC vaccine was GMP-manufactured by Eurofins CDMO (Ghent, Belgium) as a frozen suspension of bacteria in Dulbecco’s phosphate-buffered saline (DPBS) with 10% (*w/w*) PEG-6000 with a recovery of live bacteria of 18%. The vaccine was diluted in pre-manufactured, licensed sterile 0.9% saline solution certified for medicinal use. Vaccine doses were freshly prepared by dilution in a final volume of 30 mL in the pharmacy of the clinical trial site and administrated within 6 h of preparation. The dosing was based on viable colony forming units (CFU) counts. The vaccine doses were confirmed after each preparation by serial plating on tryptic soy agar and quantifying CFUs. All subjects fasted for minimum 90 min before receiving 120 mL of a bicarbonate solution (2 g in 150 mL sterile water, 1.33% *w/v*) to neutralize gastric acidity, 5 min prior to receiving 30 mL of the respective dose of ShigETEC or placebo. As placebo a saline solution with corn starch to match turbidity of the respective vaccine dose was given in the same volume as the vaccine. To guarantee blinding of the site staff, the administration was performed by an unblinded investigator and participants were asked to not discuss the taste or smell with other study participants or investigators. Evaluations and analysis were performed by blinded staff.

Safety evaluation: The safety reporting window was defined as initiating on the day of immunization through 60 days following the last dose of vaccine/placebo received by a participant. Acute Safety was defined as specific events associated with acute gastrointestinal and systemic illness symptoms on days 1 to 6 following immunization (s) including nausea, vomiting, diarrhea, abdominal pain, fever, muscle/joint aches, fatigue/malaise, headache, loss of appetite. Reactogenicity was defined as the number and proportion of expected acute safety period signs and symptom along with their associated toxicity grades. Reactogenicity events in Stage 1 were recorded by site personnel during the inpatient phase of the study. Reactogenicity events during the outpatient phase and in Stage 2 were recorded on a reminder diary card by participants at their primary residence (outpatient) and reviewed on Day 10 with study staff for recording in the database. Unexpected events during the acute safety review were recorded as Adverse Events (AEs).

Vaccine shedding analysis: For shedding analysis, stool samples were provided by the subjects as available with the target of daily sampling. Stool samples were used freshly or stored up to 48 h at 4 °C. After weighing, solid stool samples were suspended in sterile DPBS (without calcium and magnesium), liquid samples were used as they were produced. Debris were removed by centrifugation and the supernatant was used for CFU determination by plating serial dilutions onto Hektoen agar plates (BioMerieux, Marcy l’Etoile, France) in triplicates. Plates were incubated at 37 °C overnight and the colony count was determined visually for red colonies (representing commensal *E. coli* and other commensals fermenting lactose, sucrose and/or salicin) as well as for green colonies representing lactose-negative bacteria including ShigETEC. Green colonies (three representative colonies from each colony morphology types) were identified using MALDI-TOF (Bruker Microflex, Bruker, Bremen, Germany) to exclude non-*E. coli* lactose negatives (e.g., Morganella, Hafnia, etc species). If the colonies were identified as *E. coli*/*Shigella*, their identity as ShigETEC was confirmed by PCR as described below. CFU of ShigETEC was determined based on green colony count. In pilot experiments with 1 × 10^5^ CFU of ShigETEC per gram feces spiked in, 2 × 10^4^ CFU were reproducibly recovered. In volunteer experiments the detection limit was found to be heavily influenced by the abundance of the normal microbiota fermenting the carbohydrates in Hektoen agar. Naive stools from each volunteer were similarly processed to exclude presence of *Shigella*/*Salmonella* and to assess the pre-treatment microbial status. For PCR, DNA was extracted using 700 µL of the stool suspensions with QIAamp FAST DNA Stool Kit (Qiagen, Hilden, Germany) according to the manufacturer’s recommendations. DNA quality was confirmed by a 16S rDNA PCR (forward primer: 16S-Fw 3′-AGAGTTTGATCATGGCTCAG-5′; reverse primer 16S-Rev 3′-GGACTACCAGGGTATCTAAT-5′) using a Phusion High-fidelity polymerase (Thermo Fisher Scientific, Waltham, MA, USA). The reaction mix contained 5 nmol of each primer. PCR was performed in an Applied Biosystems 2720 thermocycler (Thermo Fisher Scientific) in 30 cycles of 98 °C 5 s, 58 °C 10 s and 72 °C 1 min. ShigETEC was detected using primers unique to ShigETEC with the same reaction mixture and conditions. Amplimers were detected by agarose gel electrophoresis. The detection limit of the PCR assay was found to be 10^5^ CFU/g stool using spiked-in samples. Similar to culture, abundance of the normal microbiota modified the detection limit. This PCR strategy was used to confirm ShigETEC in stool cultures with green colonies identified as *E. coli*/*Shigella*.

Immunogenicity evaluation: Blood for immunological analysis was collected on day 0 (pre-immunization) and 6, 10, 28 and 60 days after the last immunization. Native (coagulated) blood was separated by centrifugation to obtain serum that was aliquoted and frozen at −80°C at the clinical site. For antibodies in lymphocyte supernatant (ALS) generation, blood was collected into BD Vacutainer CPT separator tubes (Becton Dickinson, Franklin Lakes, NJ, USA) on day 0 (pre-immunization), and 6 and 10 days after the last immunization. Peripheral blood mononuclear cells (PBMCs) were separated by centrifugation according to the manufacturer’s instructions, cells were washed and frozen in cryoprotective medium. PBMCs were thawed and cultured at 1 × 10^7^ cells/mL in 24-well tissue culture plates for 72 h. The supernatant was collected, aliquoted and stored at −80°C until analysis. Stool for immunological analysis was collected on day 0 (pre-immunization), as well as 10 days and 28 days after the last immunization and frozen at −80°C. Stool extracts were generated from frozen stool samples by homogenizing 100 mg stool per 1 mL buffer. After centrifugation, supernatants were mixed with protease inhibitor cocktail and centrifuged again. Supernatants were collected and frozen at −80°C until analysis. Frozen serum, PBMC and stool samples were transferred to Eurofins ADME Bioanalyses (Vergèze, France) for ALS and stool extract preparation and immunogenicity analysis in qualified ELISA assay.

For the determination of vaccine-induced IgG and IgA antibodies, serum samples, ALS or fecal extracts were analyzed for the titration of specific IgG and IgA antibodies against three antigens: ShigETEC lysate, heat-stable toxin (ST, biotinylated synthetic peptide, PepScan, Lelystad, Netherlands) and heat-labile toxin B subunit (LTB, Sigma-Aldrich). Briefly, 96-well ELISA plates or Streptawell ELISA plates were coated with the respective antigen. Serial dilutions of clinical samples were run together with positive and negative controls determined in the assay qualification on each plate. Goat F(ab’)2 anti-human IgG-HRP (SouthernBiotech, Birmingham, AL, USA) and goat anti-human IgA-HRP (SouthernBiotech) were used for IgG and IgA, respectively, and detected with Sureblue TMB, blocked with H_2_SO_4_, and OD_450_ was measured with a Spectramax i3x plate reader (Mesoscale Discovery). The determined titer corresponds to the highest dilution factor giving a signal value above the mean signal of the negative control for serum and stool extracts. For stool extracts, total IgA was measured, and titers calculated in ratio of total IgA content. For ALS, titers were determined as the highest dilution factor giving a signal over two times the value of the average technical background signal.

LTB neutralization assay: The neutralizing capacity of anti-LTB antibodies in serum was assessed in an LTB-GM1 binding assay. Serum dilutions were incubated with 0.5 ng LTB (Sigma-Aldrich), and the amount of LTB that remained free from antibody binding was quantified by binding to GM1-coated plates by ELISA using an anti-cholera toxin beta antibody (Sigma-Aldrich) for detection.

Statistical considerations: The primary objective of the study was to assess safety and tolerability, and sample size considerations are based on prevalence of uncommon AEs. Given that 56 subjects in the trial did receive one or more doses of ShigETEC vaccine, there is approximately a 25% chance of observing at least one instance of an uncommon AE to occur in only 1 in 100 vaccinated subjects. Chances are 75% of observing a more common AE expected to occur in 1 in every 25 vaccinated subjects.

## 3. Results

### 3.1. Participants

In Stage 1 of the Phase 1 clinical trial, 68 subjects were screened of which 48 subjects were randomized and sequentially enrolled in 4 ascending dose groups (12 subjects per group) to receive a single oral dose of the ShigETEC vaccine starting in the first group, 1A, at a dose, of 1 × 10^9^ Colony Forming Units (CFU) of vaccine or a placebo, followed by 1 × 10^10^ CFU, 5 × 10^10^ CFU and finally 2 × 10^11^ CFU per dose in groups 1B, 1C and 1D, respectively. In Stage 2, 59 subjects were screened from which 36 subjects were randomized and sequentially enrolled in 3 groups receiving 2 (2A), 3 (2B) or 4 (2C) vaccinations with ShigETEC. All participants in the study were white and one was Hispanic/Latino. The age and gender distribution of participants is detailed in Table 1. The average age in Stage 1 was 30.2 years, in Stage 2 31.6 years. In total, out of the 84 participants who received vaccine/placebo there were 30 females and 54 males. In Stage 1 a gender bias was noted in the distribution of male and female subjects. The clinical site was advised to include more female participants in Stage 2 if possible, and consequently, the gender distribution was more even in Stage 2 (Table 1).

### 3.2. Single Ascending Dose Administration

#### 3.2.1. Safety

In Stage 1, 48 subjects were included in 4 ascending dose cohorts. In total six adverse events and three reactogenicity events occurred, none of which qualified as severe. Mildly increased respiration rates frequently occurred in all cohorts, none showing any correlation with cohort or group assignment. The six non-reactogenicity type events occurred in 4 patients with three events being unrelated to the vaccine. Two subjects reported mild headaches–one subject on day 5 and the other subject on day 1 and day 3, which are possibly related to vaccination since both were in the vaccine group.

Three reactogenicity events were reported in two vaccinees and occurred on the vaccination day all in cohort 1D, the highest dose group. These events were vomiting and diarrhea, both graded as mild, and one event of nausea graded as moderate (Table 2). The reported events were unrelated to the onset, frequency, or duration of observed vaccine shedding. No reactogenicity type events occurred in any of the other groups. No reported event qualified as severe. Since reactogenicity events were reported in the highest dose group, the dose for the multi-dose regimen was chosen at the second highest dose tested, namely 5 × 10^10^ CFU.

#### 3.2.2. Vaccine Shedding

Shedding of the vaccine strain in stool after single dose administration of ShigETEC was only seen in vaccine recipients and was of short duration following dosing with a maximum of 3 days after dosing (Figure 2). Based on the observed duration of shedding of vaccine the strain, the dosing interval for the multiple dosing regimen in Stage 2 was set at 3 days.

### 3.3. Multiple Dose Administration

Based on the safety, tolerability and shedding of the vaccine observed in Stage 1, the dose for the multidose regiment in Stage 2 was chosen at 5 × 10^10^ CFU. Increasing number of doses, namely 2, 3 and 4 vaccinations, given 3 days apart to identify the optimal vaccination regimen. Stage 2 included 3 cohorts with 12 subjects each with a placebo to vaccine ratio of 1:2.

#### 3.3.1. Safety

In Stage 2, the vaccine was well tolerated across the three multidose regimens (Table 2). In the two-dose group, one out of 8 vaccine recipients reported nausea on the day of the second vaccination. In the three-dose group a total five reactogenicity events were reported in 2 subjects, all on vaccination days (day 1). One participant vomited on day 1 and another participant reported nausea on day 4 as well as on day 7 and vomiting on day 1 and day 7. All events were graded as mild. In the four-dose group only one participant experienced reactogenicity events, reporting vomiting on the vaccination days of all four doses and having diarrhea one day after the last vaccination. All events were graded as mild. All reactogenicity events in Stage 2, similarly to Stage 1, resolved within one day and were unrelated to the onset, frequency, or duration of observed vaccine shedding. No reactogenicity event occurred in the placebo groups. Five non-reactogenicity adverse events were reported in Stage 2, all events were assessed as mild and not related to the vaccine.

#### 3.3.2. Vaccine Shedding

Shedding of the ShigETEC vaccine was observed in 22 of the 24 vaccinees in the multi-dose regimen and increased in duration with the number of doses (Figure 3). In the two-dose group, shedding was detected in 7 out of 8 vaccinees and lasted one to maximum 2 days after the last vaccination. In the three-dose group, vaccine shedding was observed in all 8 subjects who received ShigETEC with half of the group shedding until at least two days after the last vaccination. In the four-dose group, vaccine shedding was reported in 7 out of 8 vaccinees with 3 subjects shedding until two days after the last vaccination. In each of these groups, one subject shed vaccine until 3 days, and one until 5 days after the last immunization.

### 3.4. Immunogenicity of Oral Vaccination with ShigETEC

Analysis of vaccine-specific immune responses in serum using ShigETEC lysate as antigen, revealed a robust and strong IgA response (Figure 4A). Even with single immunization with the highest two doses (cohort 1C and 1D) a strong anti-ShigETEC IgA response was observed in most of the vaccinees, with 7 out of 8 vaccinees (87,5%) showing an at least 4-fold titer increase. The number of responders further increased (to 100%) upon multiple immunizations with ShigETEC, and 100% of vaccinees showed at least 4-fold increase in anti-ShigETEC IgA responses after 4 vaccinations. The geometric mean titers were comparable in the 2, 3 and 4-times immunized cohorts and higher than in the 1-time vaccinated group (Figure 4B). However, superiority of the 4-times vaccination schedule was revealed by the comparison of fold increases in anti- ShigETEC IgA geometric mean titer (GMT) values (Figure 4C, Table A1). Analyzing the serum samples for ShigETEC-specific IgG showed much lower levels in terms of titer increases and number of responders, with no vaccinees showing a titer increase of 4-fold or higher. No specific anti-ShigETEC titer increase was detected in placebo recipients.

Analysis of ALS samples for IgA antibodies against the ShigETEC lysate detected responses only among the vaccinees and not among placebo recipients (Figure 5A). The number of responders as well as the level of the responses show correlation with the number of immunizations up to 3 doses but seems to be inferior in the 4-dose regimen. Since the samples analyzed were collected at days 6 and 10 following the last booster immunization, which correspond to days 16 and 20 in the 4-dose cohort, it can be speculated that at these timepoints the number of circulating antibody-producing lymphocytes declines already.

Anti-ShigETEC IgA responses in stool were observed in several subjects in each cohort (Figure 5B). Notably, anti-ShigETEC IgA titer increases correlated between serum and fecal extracts of individuals (Figure 5C), suggesting that serum IgA titers reflect the mucosal response to the vaccine strain.

In case of the LTB antigen, an opposite tendency was observed compared to the anti-ShigETEC lysate response in terms of Ig class. While there was a good serum anti-LTB IgG response, the IgA response was much weaker (Figure 6, Table A1 and Table A2). 50% of vaccinees in the 4-times immunized cohort mounted anti-LTB IgG titers, which were at least 4-fold increased over pre-immunization levels, while no such responses were detected in the other cohorts. The individual titers for all subjects in the 4-dose group are shown in (Figure 6B). Four sera from the 4-times vaccinated group showed a vaccination-induced increased LTB neutralizing capacity based on blocking of LTB binding to ganglioside GM1 (Figure 6C), which strongly correlated with the corresponding serum anti-LTB IgG titer. In case of the ST antigen, no individuals mounted a 4-fold titer increase in any cohort at any timepoints investigated, however, in the 4-dose group, 3 out of 8 vaccinees had a 2-fold increase in anti-ST IgG titer.

The anti-LTB response observed in serum was also reflected in stool extracts with 3 out of the 4 responders who showed a more than 4-fold increase in serum anti-LTB IgG also showing a more than 4-fold increase in anti-LTB IgA in fecal extracts. In general, the data quality from stool extracts was poorer based on low total IgA levels. For analysis of ALS samples against the LTB, IgA responses were detected in some individuals, however, most likely due to the late sampling time point for circulating antibody producing lymphocytes, in the 4-dose group most responses were likely missed.

## 4. Discussion

The ShigETEC live oral vaccine was well-tolerated in this first-in-man study. Gastrointestinal reactogenicity events (nausea, vomiting, diarrhea) occurred in 4 of the 24 subjects (~17%) receiving 5 × 10^10^ CFU dose, independent of the number of immunizations, all mild or moderate, and self-resolving within a day. No reactogenicity adverse events occurred in the placebo groups. There is a historic precedent for such good tolerability of high vaccine doses with a previous live attenuated *Shigella* vaccine, Vadizen, which was developed in Romania in the 1970′s and proved to be efficacious in field studies involving thousands of children and adults [15]. There are several parallels between ShigETEC and Vadizen, which was based on the Istrati-32 *S. flexneri* 2a strain that was also non-invasive due to a large deletion in the invasion plasmid that inactivated the Type3 Secretion System (T3SS), essential for invasiveness [16]. Notably, vaccine shedding with Vadizen was also on average 3 days. The field studies were performed with 5-times oral vaccination with 2 × 10^11^ CFUs. The Phase 1 study design of ShigETEC was built on the expectations learned with this previous vaccine. However, immunogenicity data from studies with Vadizen are not available. For ShigETEC, we detected strong IgA response against the vaccine strain (used as whole lysate preparation for ELISA) even after single oral immunization with the 5 × 10^10^ and 2 × 10^11^ CFU doses. The vaccine-induced increases in IgG titers were moderate, which could be explained by the already high IgG levels against the ShigETEC lysate in pre-vaccination samples which could originate from cross-reactive anti-*E. coli* antibodies due to the high similarity between *Shigella* and *E. coli*. Alternatively, the mucosal route of administration may explain the higher level of IgA response. On the contrary, higher IgG vs. IgA titer increases were found against LTB. This may be related to the nature of this antigen and/or the known adjuvant effect of LTB that triggers preferably this kind of isotype switch. When 4 doses of ShigETEC vaccine were administered, 50% of vaccinees showed an at least 4-fold increase in serum anti-LTB IgG titer. The anti-LTB antibodies were found to be functional, i.e., were able to neutralize LTB binding to its cognate receptor. The anti-LTB serum IgG responses induced in subjects after vaccination with ShigETEC proves that the LTB-ST fusion construct in ShigETEC was indeed expressed, immunogenic and was an adequate antigen for the generation of neutralizing antibodies. Since we observed a very strong correlation between anti-LTB serum IgG titers and LT-neutralizing capacity, monitoring such serum IgG antibodies could be considered and investigated as a potential correlate of protection against ETEC toxin-mediated diarrhea in future studies with ShigETEC.

Neither the frequency nor the duration of observed vaccine shedding correlated with the strength of vaccine-induced immune responses. Some subjects, in whom no shedding could be detected, still mounted high vaccine-specific immune responses detectable in serum, which correlated with mucosal responses. Thus, the detection of vaccine shedding does not predict the effectiveness of the vaccination with ShigETEC to induce an immune response. The reported vomiting in 4 subjects, which never occurred immediately after ingestion of the vaccine but several hours later, did not correlate with poor immunogenicity, indicating that the ShigETEC vaccine was sufficiently ingested and not lost due to these reactogenicity events.

The LT-neutralizing antibody response to the vaccine observed in the different cohorts clearly indicates the superiority of multiple vaccine doses. While lower number of immunizations also elicited a strong and significant anti-ShigETEC IgA response, the 4-times vaccination regimen resulted in the highest number of responders and highest level of antibody responses overall. Consequently, this vaccination regimen appears to be the most attractive for future clinical studies aiming at determining vaccine mediated protection in human challenge studies.

While the antibody responses to the ST were more modest compared to those against *Shigella* and LTB, it represents the first reported successful generation of anti-ST serum antibodies with an oral live attenuated vaccine. The sharp increase in response rate and level of antibodies to the LT and ST toxins between 3 and 4 immunization groups suggest that a 5-time dose regimen is likely to be beneficial to achieve an even higher anti-ETEC immunity.

## 5. Conclusions

The study showed that the oral ShigETEC vaccine was well tolerated at doses up to 5 × 10^10^ CFU/dose at up to 4 doses at 3-day intervals with only minor and transient reactogenicity noted. The vaccine will proceed to additional Phase 2 testing that will include evaluation of protection in controlled human infection (CHIM) models.

## 6. Patents

Eveliqure Biotechnologies GmbH holds patents and patent applications related to this project.

## Figures and Tables

**Figure 1 vaccines-10-00340-f001:**
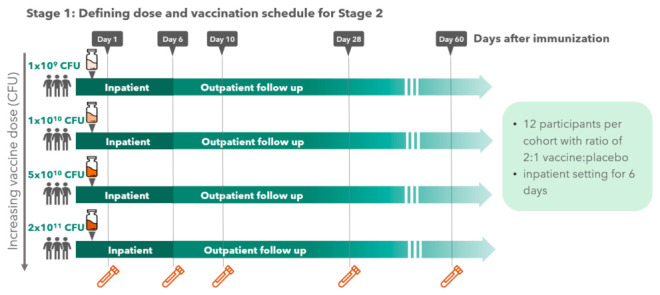
Study design.

**Figure 2 vaccines-10-00340-f002:**
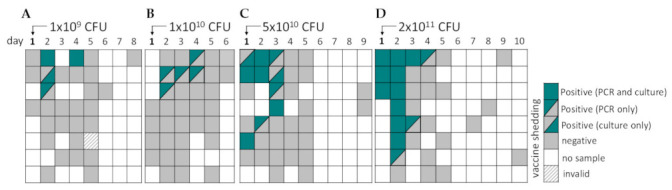
Vaccine shedding in stool after single oral vaccination with ShigETEC. Stool samples were obtained as available from subjects who received a single oral immunization with (**A**) 1 × 10^9^ CFU, (**B**) 1 × 10^10^ CFU, (**C**) 5 × 10^10^ of (**D**) 2 × 10^11^ CFU ShigETEC and assessed for the presence of vaccine strain by PCR and culturing. PCR was considered invalid, when two independent 16S rDNA PCR assays yielded negative results and repeated DNA extraction was not possible due to small sample amount. Each line corresponds to one subject.

**Figure 3 vaccines-10-00340-f003:**
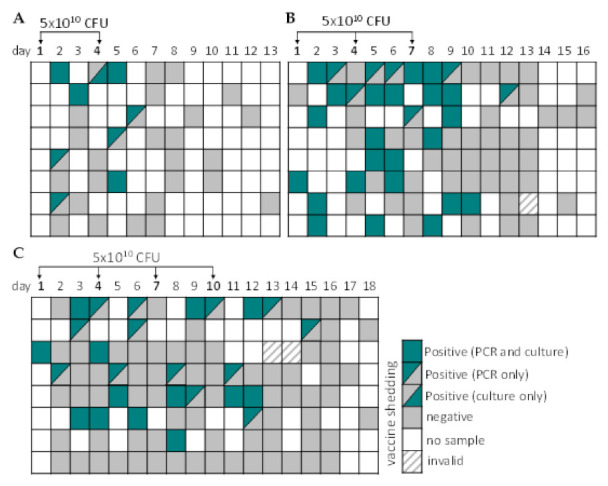
Vaccine shedding in stool after multiple oral vaccinations with ShigETEC. Stool samples were obtained as available from subjects who received (**A**) two, (**B**) three or (**C**) 4 doses of 5 × 10^10^ CFU ShigETEC and were assessed for the presence of vaccine strain by PCR and culturing. PCR was considered invalid, when two independent 16S rDNA PCR assays yielded negative results and repeated DNA extraction was not possible due to small sample amount. Each line corresponds to one subject.

**Figure 4 vaccines-10-00340-f004:**
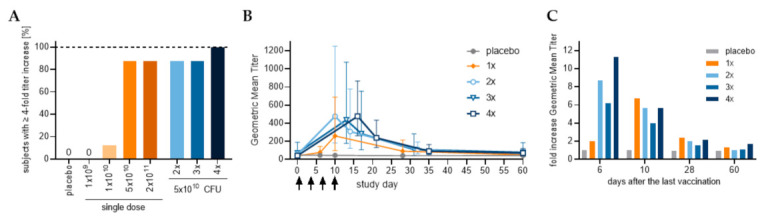
Serum anti-ShigETEC lysate responses. (**A**) Subjects responding with at least 4-fold increase in anti-ShigETEC IgA titer at any sampling day compared to pre-immune titer in serum. (**B**) Geometric mean titers and % CV of serum anti-ShigETEC IgA are given for all cohorts which received 5 × 10^10^ CFU/dose ShigETEC or placebo. (**C**) Fold increase of geometric mean titers for all cohorts which received 5 × 10^10^ CFU/dose ShigETEC or placebo.

**Figure 5 vaccines-10-00340-f005:**
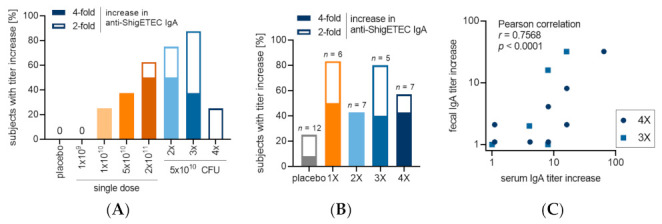
Anti-ShigETEC IgA responses in ALS and stool extracts. (**A**) anti-ShigETEC IgA responders in ALS (**B**) anti-ShigETEC IgA responders in stool extracts from available samples with sufficient total IgA content to allow comparison of samples collected at different time points from the same subject (**C**) Pearson correlation between fecal and serum IgA titer increases detected by ELISA using ShigETEC lysate as antigen with samples of individuals in the three-dose (square) and four-dose (circle) groups.

**Figure 6 vaccines-10-00340-f006:**
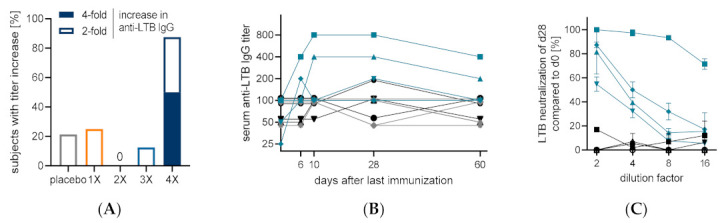
Serum anti-LTB IgG responses and LTB neutralization capacity. (**A**) Percentage of subjects responding with an at least 4-fold (solid) or at least 2-fold (open) titer increase in anti-LTB IgG in serum at any analyzed time point in groups receiving indicated numbers of doses of 5 × 10^10^ CFU oral ShigETEC or placebo (grey). (**B**) Individual anti-LTB serum IgG titers of all subjects in group 2C (4-dose regimen) with placebo recipient in grey, vaccinees with <4-fold titer increase in black and vaccinees with >4-fold titer increase in blue. (**C**) LTB neutralizing capacity of serum of all subjects of group 2C (4-dose regimen) at indicated dilution factor.

**Table 1 vaccines-10-00340-t001:** Age and gender distribution by cohort and group assignment.

		Average Age [Years]	Gender [% Females]
Stage 1 Single dose	Cohort 1A	30.9	16.7
	Cohort 1B	33.4	8.3
	Cohort 1C	28.0	33.3
	Cohort 1D	28.3	58.3
	Total	30.2	29.2
Stage 2 Multiple doses	Cohort 2A	32.3	41.7
	Cohort 2B	29.3	41.7
	Cohort 2C	33.0	50.0
	Total	31.6	44.4
Stage 1 Single dose	Placebo	30.8	18.8
	Vaccine	29.8	34.4
	Total	30.2	29.2
Stage 2 Multiple doses	Placebo	33.3	41.7
	Vaccine	29.3	45.8
	Total	31.6	44.4

**Table 2 vaccines-10-00340-t002:** Reported reactogenicity events in vaccine recipients.

Stage	Cohort	Number of Subjects with Reactogeni-City Events	Reacto-Genicity Event	Study Day	Number of Vaccinees (%)	Number of Events	Severity (Grade)
Stage 1	1D	2	Diarrhea	D1	1 (12.5%)	1	Mild
			Nausea	D1	1 (12.5%)	1	Moderate
			Vomiting	D1	1 (12.5%)	1	Mild
Stage 2	2A	1	Nausea	D4	1 (12.5%)	1	Mild
	2B	2	Nausea	D4	1 (12.5%)	1	Mild
				D7	1 (12.5%)	1	Mild
			Vomiting	D1	2 (25%)	2	Mild
				D7	1 (12.5%)	1	Mild
	2C	1	Diarrhea	D11	1 (12.5%)	1	Mild
			Vomiting	D1	1 (12.5%)	1	Mild
				D4	1 (12.5%)	1	Mild
				D7	1 (12.5%)	1	Mild
				D10	1 (12.5%)	1	Mild

## Data Availability

Not applicable.

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
