# Peer review of "Evaluation of the Safety, Tolerability and Immunogenicity of ShigETEC, an Oral Live Attenuated Shigella-ETEC Vaccine in Placebo-Controlled Randomized Phase 1 Trial"

_vaccines, 2022, doi:10.3390/vaccines10020340_

Round 1
Reviewer 1 Report
This manuscript describes the safety, tolerability and immunogenicity of the ShigETEC vaccine. It was well-designed with a dose/dosing escalation to determine MTD.
- For the highest single dose group (D), was the shedding on day 1 in those with reactogenicity response or were they unrelated?
- It may be a function of group size, but it is noted that only 1 participant in both the two dose and four dose regimen experienced adverse reactogenicity while 5 of 8 did in the three dose group? Is the difference in the ALS response noted between the three dose and four dose groups any relation to that?
- What is meant by "usable data" for the ALS response?
- in the discussion, it is noted that the moderate IgG titer increases may originate from masking effect of high levels of baseline anti-E.coli IgG. Was this measured or observed?
- There is alot of discussion of the immune response in terms of serum antibody response. It is not entirely surprising that there wasn't a correlate with serum response and vaccine shedding - it would be expected that only mucosal response would correlate. This warrants further investigation.
Reviewer 2 Report
The manuscript by Girardi et al is a phase 1 human trial study of the ShigETEC live vaccine. The need of a vaccine to protect populations against Shigella and E. coli is of much significance. Overall, the study appears to be performed well. However, some areas to better help present this study are listed below along with some minor grammatical edits.
Line 30. Define ALS
Line 49. Should this be approved rather than registered?
Lines 50-56 & 70-75. It would seem appropriate to include some general references to the pathology/ virulence of Shigella and ETEC in the respective sections.
Study Design: I would recommend including a table to outline details of the trials. It was not clear # of doses and intervals between Stage 1 & 2 in this section, especially based upon sentence 121-122.
Line 107. Define what the "stopping rules" were for this study.
Line 127-129. Reference for this statement.
Lines 156-160. Volunteers received the vaccine in 120 ml. Placebo group[ received 30 ml. But following sentence reads ".... dose was given in same volume....." Is something incorrect here?
Table 2. Column 1. Formatting is distorted for label.
Figure 1. Using larger font for legend could be useful. Hard to read in printed version of the manuscript.
Discussion. For the authors' consideration. For those volunteers that had some level of reactogenicity to the vaccine (vomitting), was there any correlation poor immune response or vaccine retention since they potentially lost the vaccine itself? Just curious.
